# Prediction of Cobb Angle Using Deep Learning Algorithm with Three-Dimensional Depth Sensor Considering the Influence of Garment in Idiopathic Scoliosis

**DOI:** 10.3390/jcm12020499

**Published:** 2023-01-07

**Authors:** Yoko Ishikawa, Terufumi Kokabu, Katsuhisa Yamada, Yuichiro Abe, Hiroyuki Tachi, Hisataka Suzuki, Takashi Ohnishi, Tsutomu Endo, Daisuke Ukeba, Katsuro Ura, Masahiko Takahata, Norimasa Iwasaki, Hideki Sudo

**Affiliations:** 1Department of Orthopaedic Surgery, Hokkaido University Hospital, N15W7, Sapporo 060-8638, Hokkaido, Japan; 2Department of Orthopaedic Surgery, Eniwa Hospital, 2-1-1 Kogane-chuo, Eniwa 061-1449, Hokkaido, Japan; 3Department of Advanced Medicine for Spine and Spinal Cord Disorders, Faculty of Medicine and Graduate School of Medicine, Hokkaido University, N15W7, Sapporo 060-8638, Hokkaido, Japan

**Keywords:** adolescent idiopathic scoliosis, deep learning algorithm, three-dimensional depth sensor, underwear

## Abstract

Adolescent idiopathic scoliosis (AIS) is the most common pediatric spinal deformity. Early detection of deformity and timely intervention, such as brace treatment, can help inhibit progressive changes. A three-dimensional (3D) depth-sensor imaging system with a convolutional neural network was previously developed to predict the Cobb angle. The purpose of the present study was to (1) evaluate the performance of the deep learning algorithm (DLA) in predicting the Cobb angle and (2) assess the predictive ability depending on the presence or absence of clothing in a prospective analysis. We included 100 subjects with suspected AIS. The correlation coefficient between the actual and predicted Cobb angles was 0.87, and the mean absolute error and root mean square error were 4.7° and 6.0°, respectively, for Adam’s forward bending without underwear. There were no significant differences in the correlation coefficients between the groups with and without underwear in the forward-bending posture. The performance of the DLA with a 3D depth sensor was validated using an independent external validation dataset. Because the psychological burden of children and adolescents on naked body imaging is an unignorable problem, scoliosis examination with underwear is a valuable alternative in clinics or schools.

## 1. Introduction

Adolescent idiopathic scoliosis (AIS) is the most common form of pediatric spinal deformity. Early detection of deformity and timely intervention, such as brace treatment, will contribute to the inhibition of progressive changes. Adam’s forward bend test using a scoliometer is a representative method for detecting scoliosis. Although it provides relatively high sensitivity (83.3%) and specificity (86.8%) in scoliosis detection [1,2], the correlation coefficient between the scoliometer measurement and Cobb angle in standing radiographs is not satisfactory (r = 0.677) [3].

A newly developed class II medical device named Scoliomap is commercially available in Japan. Scoliomap comprises a three-dimensional (3D) depth sensor and an algorithm installed on a laptop computer, providing a predicted Cobb angle and color deviation map within 1.5 s of shooting the surface of the back [4,5]. Recently, a 3D depth sensor imaging system was modified with a newly developed convolutional neural network (CNN) [6]. Although the recent system exhibited a significant ability to predict the Cobb angle, a major limitation of the previous study was that an external validation dataset could not be prepared [6]. The performance of the deep learning algorithm (DLA) should be assessed using an independent external validation dataset [6]. In addition, while the 3D depth sensor imaging system with CNN is expected to be used for screening scoliosis in clinics or physical examination at schools, the psychological burden of children and adolescents on naked body imaging is an unignorable problem.

To address these shortcomings, in this study, we evaluated the performance of the DLA in predicting Cobb angle using an independent external validation dataset and assessed the predictive ability depending on the presence or absence of clothing in the prospective analysis.

## 2. Materials and Methods

### 2.1. Subjects

The institutional review board of Hokkaido University Hospital approved the experiments and included any relevant details. Informed consent for this study and publication of the information/case study was obtained from all participants and/or their guardians/parents. This study was conducted at two scoliosis centers in Japan from November 2021 to September 2022. One hundred patients with suspected scoliosis who were independent of subjects in previous studies [4,5,6] were collected. The inclusion and exclusion criteria were the same as those of previous studies [4,5,6]. The inclusion criteria were as follows: (i) Aged 7 to 18 years old, (ii) referral with a confirmed diagnosis based on X-rays, (iii) no history of brace treatment, and (iv) written informed consent. Patients with symptomatic neuromuscular or congenital scoliosis were excluded. Information about age, gender, and the history of brace treatment was obtained from medical records. Cobb angles were measured three times by a spine specialist, and the average angles were used [5]. The Cobb angles on standing radiographs were measured without knowledge of the predicted results of the DLAs. All steps were completed according to the relevant guidelines and regulations.

### 2.2. Prediction of Cobb Angle Using DLA

The DLAs for the prediction of the Cobb angle are shown in Figure 1. In brief, DLAs to predict the Cobb angle involve the following steps:

#### 2.2.1. 3D Depth Sensor Imaging

The system comprises a 3D depth sensor (Xtion Pro Live, ASUSTeK Computer Inc., Taipei, Republic of China) and a laptop computer (Core-i5, 7200U-4 GB HP pavilion-15-au105tu, HP Inc., Palo Alto, CA, USA) [4,5,6]. A 3D point cloud of the subject’s back surface was scanned by the sensor. A 3D point cloud, P1, was obtained.

#### 2.2.2. Estimation of Approximated Median Sagittal Plane and Region of Interest

The pose-normalized point cloud P2 from P1 was obtained using principal component analysis in which the approximated median sagittal plane was arranged. The region of interest (ROI), which comprises a square box generated from the waistline to both shoulders, was obtained as P3 [4,5,6].

#### 2.2.3. Generation of Reflected Point Clouds

The reflection point group P3r as the mirror projection of P3 in relation to the sagittal plane was generated [4,5,6].

#### 2.2.4. Conversion to Input Data for Predicting Cobb Angle by DLAs

The difference in the position of the original P3r relative to the original P3 was converted to a Comma Separated Value (CSV) comprising 159 rows and columns, where zeros were filled into boxes without numbers [6].

#### 2.2.5. Prediction of Cobb Angle by DLAs

The architecture of the CNN model is shown in Figure 1. The CNN models comprised 14 layers, including a combination of a max-pooling layer and three convolution layers. In a previous study, five-fold cross-validation with ten repeats was performed on 160 CSV files to create trained DLA models, from which 50 trained DLAs were obtained [6]. The average predicted Cobb angle derived from the 50 trained DLAs was calculated.

To predict the Cobb angle using 50 trained DLAs, we used a computer equipped with a central processing unit of Core i7-9750H (Intel), a graphics processing unit of GeForce RTX 2070 (NVIDIA), and random-access memory of 32GB.

### 2.3. Shooting Patterns

The back surfaces of the subjects were shot under the following four shooting patterns (Figure 2): (1) Adam’s forward bending without underwear, (2) Adam’s forward bending with underwear, (3) standing posture without underwear, and (4) standing posture with underwear.

### 2.4. Pilot Study Using Phantom Models

To evaluate the performance of the DLA in predicting Cobb angle using three different underwear types and two colors, phantom models were obtained from plaster wrap castings to make hard braces for single thoracic curves, single thoracolumbar/lumbar curves, and double thoracic and thoracolumbar/lumbar curves (Figure 3). The back side of each phantom model with each underwear was scanned 10 times. The following underwear types were assessed: (a) AIRism (UNIQLO CO., LTD, Yamaguchi, Japan), (b) Women’s Side Seamless Jersey Crew Neck T-Shirt (Ryohin Keikaku Co., Ltd., Tokyo, Japan), and (c) Nameraka COTTON (SHIMAMURA Co., Ltd., Saitama, Japan).

### 2.5. Statistical Analysis

In the pilot study, the one-way ANOVA was applied to assess the difference in predicted Cobb angles among three brand underwear types or between white and black. Repeatability was assessed in each group using the coefficient of variation (CV), which was calculated by dividing the standard deviations of intrasubject results by the mean and multiplying by 100 to present the result as a percentage [7]. A CV of less than 10% was considered as very good, 10% < CV ≤ 20% as good, 20% < CV ≤ 30% as fair/moderate, and CV > 30% as poor [8]. The correlation coefficient, mean absolute error (MAE), and root mean squared error (RMSE) between the actual and predicted Cobb angles were calculated. In addition, MAE and RMSE were evaluated according to curve severity: The mild group with a Cobb angle in the range of 0°–19°, which does not need treatment, the moderate group with a Cobb angle in the range of 20°–39°, which may require brace treatment, and the severe group with a Cobb angle ≥ 40°, which may be a surgical indication. Sensitivity, specificity, positive predictive value, negative predictive value, accuracy, positive likelihood ratio, and negative likelihood ratio were estimated as experimental indicators of Cobb angles of 10°, 15°, 20°, and 25°. Data analyses were performed using the JMP statistical software for Windows (version 14; SAS, Inc., Cary, NC, USA). Statistical significance was set as *p* value < 0.05.

## 3. Results

### 3.1. Pilot Study Using Phantom Models

The predicted Cobb angles were not significantly different among the three black underwear types in each curve model (Table 1). The CV was <5% for all underwear types, which represented good repeatability (Table 1). Compared with the white and black in the UNIQLO underwear, there were no significant differences between the two colors, and the CV was <4% in each curve model (Table 2). No significant differences were observed among the three brands. Considering that the skin is not transparent, and as an enterprise that is expanding overseas, the black UNIQLO underwear was selected in the following human subjects’ experiments.

### 3.2. Human Subjects

The mean age was 13.2 ± 2.3 years, and the average Cobb angle of the maximum curve was 26.0 ± 12.0° (range, 5° to 60°) on standing radiographs. The correlation coefficients are shown in Figure 4. The correlation coefficient was 0.87 in pattern 1 (Adam’s forward bending without underwear) group. There were no significant differences in the correlation coefficients among the four shooting patterns.

The MAEs and RMSEs are listed in Table 3. The MAE and RMSE were 4.7° and 6.0°, respectively, in the pattern 1 group. The *p*-values of the Tukey post hoc test between the two MAEs in each shooting pattern are listed in Table 4. There was no significant difference in the MAE between the pattern 1 and pattern 2 (Adam’s forward bending with underwear) groups (*p* = 0.99), whereas significant differences were observed between the pattern 1 and pattern 3 (standing posture without underwear) groups (*p* = 0.03). There was also a significant difference between the pattern 2 and pattern 3 groups (*p* = 0.04).

The experimental indicators of the averaged predicted Cobb angles of 10°, 15°, 20°, and 25° are listed in Table 5. At a Cobb angle of 10°, which was diagnosed as scoliosis, the accuracy of the averaged predicted Cobb angle was 0.92 in the pattern 1 group.

## 4. Discussion

The present study demonstrated that the correlation coefficient was 0.87, and the MAE and RMSE were 4.7° and 6.0°, respectively, in Adam’s forward bending without underwear group. The accuracy of detecting scoliosis with a curve ≥10° was 92%. We previously documented that the coefficient of correlation between the Scoliomap measurement and the Cobb angle was 0.85 [5]. In addition, the 3D depth sensor imaging system with its newly created CNN for regression showed the correlation between the actual and the mean predicted Cobb angles was 0.91, and the MAE and RMSE were 4.0° and 5.4°, respectively [6]. The accuracy of the mean predicted Cobb angle was 94% for identifying a Cobb angle ≥ 10° [6]. It has been reported that a DLA trained and tested for a certain category of samples may not work when it is adopted to different test samples [6,9]. However, the present study validated that our DLA with a 3D depth sensor could predict the Cobb angle with greater predictive ability compared to the Scoliomap system using an independent external validation dataset.

Although the predicted Cobb angle was not calculated, Yang et al. [10]. developed DLAs for automated scoliosis detection using unclothed 2D back photographs, showing that the accuracy for detecting scoliosis was 0.75 with a curve ≥10°. They also reported the accuracy of the DLA was lower in the external data set than in the internal data set, and the area under the curve decreased from 0.95 to 0.81 with a curve ≥10° [10]. In the present study, there was no significant difference in the predicted Cobb angles between white and black underwear types. Compared to the photograph, the numerical data captured by the 3D depth sensor has more information to detect scoliosis regarding altitude difference on the back surface, independent of recording conditions, such as indoor lighting [6].

The other reason why our DLA maintained high performance using external independent test samples may be that the input data were preprogrammed to focus on the essential part of the human body [6]. Feature selection is frequently performed in DLA to improve accuracy [11]. Although a small sample size of training datasets was used, the input data were processed as the ROI in a previous phase, which was a square box generated from the waistline to both shoulders to capture the fundamental features of the data [6].

Although the present study exhibited no significant differences in the correlation coefficient among the four groups, the correlation coefficient was the highest in Adam’s forward bending without underwear group. The correlation coefficients were almost identical between the group with and without underwear under Adam’s forward bending posture. In addition, there was no significant difference in MAE between the groups with and without underwear in the bending posture. These results indicate that Adam’s forward bending without underwear results in the highest performance in predicting the Cobb angle. However, there is no significant difference in the predicting ability with or without underwear as long as the subjects are bending forward and are wearing well-fitting underwear. As mentioned, the most valid data for the 3D depth sensor are numerical data regarding the altitude difference on back surface and the forward bending test results in the left-right difference compared to the standing posture in patients with scoliosis.

This study had some limitations. First, the subjects included children or adolescents suspected of having AIS and were referred to our hospitals, resulting in 8/100 participants having no scoliosis. This is acceptable when considering an alterable device for X-rays to monitor curve progression, precluding excess radiographs for mild cases of scoliosis [6]. However, the outcomes in the DLA with a 3D depth sensor system may be overfitted in school screening because the study setting was different from school screening, where most children have no scoliosis [5,6]. When the performance of our DLA is validated in a large-scale clinical trial targeting scoliosis school screening, this system is expected to be used for screening scoliosis [5,6]. Second, because the DLA used in this study was the same as in the previous study [6], the standing posture may have resulted in unfavorable results. Strictly, a comparison test of standing and forward bending is required using DLAs structured for each posture.

## 5. Conclusions

The performance of the DLA with a 3D depth sensor was validated for predicting the Cobb angle using an independent external validation dataset. Because the psychological burden of children and adolescents on naked body imaging is an unignorable problem, scoliosis examination with underwear is a valuable alternative in clinics or schools.

## Figures and Tables

**Figure 1 jcm-12-00499-f001:**
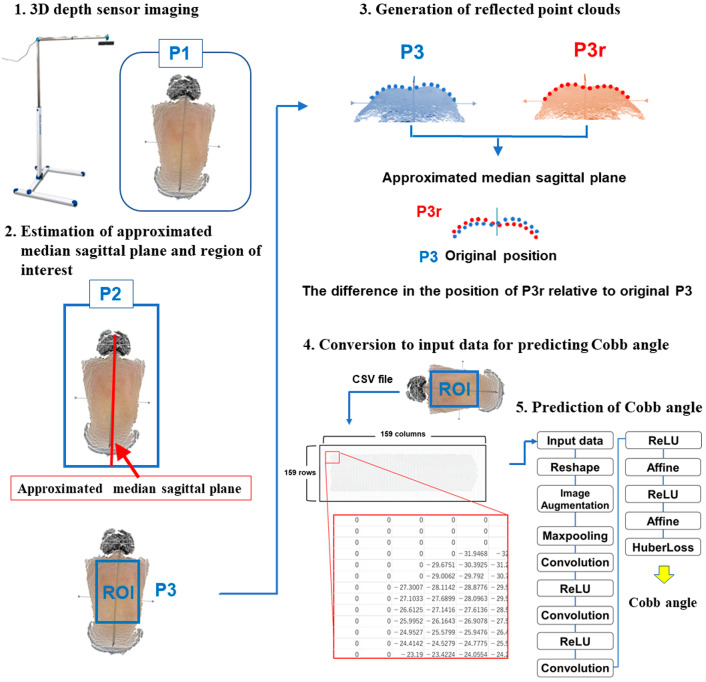
Previously developed algorithm [4,5,6] to obtain input data used in the deep learning algorithm for prediction of Cobb angle. The comma-separated value (CSV) files were generated by converting from the point clouds within the region of interest. The CSV files included 159 rows and 159 columns, where zeros were filled into boxes without a number.

**Figure 2 jcm-12-00499-f002:**
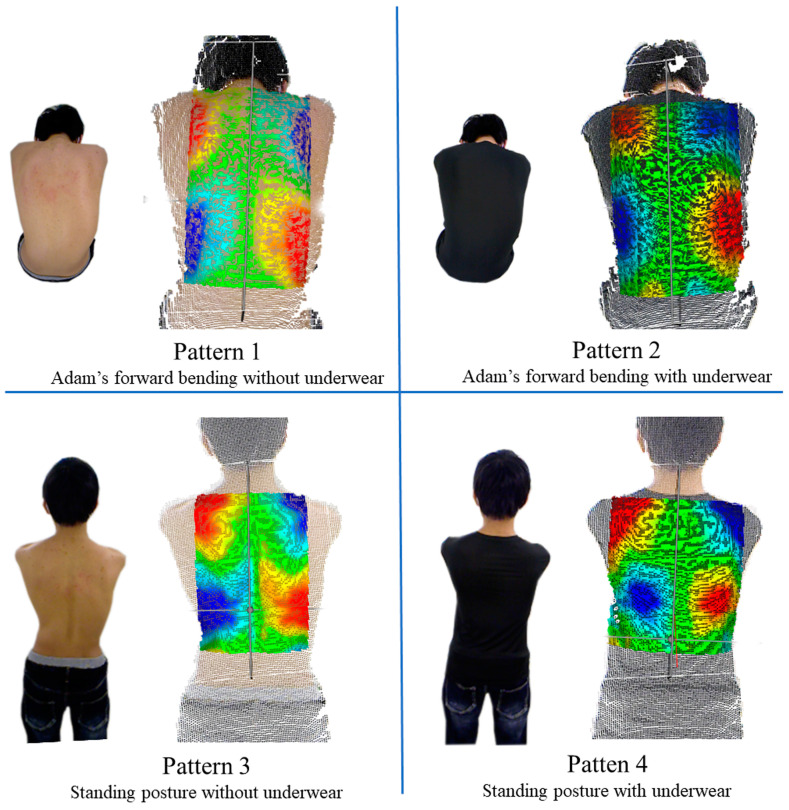
Patient’s back surfaces were shot under the following four shooting patterns.

**Figure 3 jcm-12-00499-f003:**
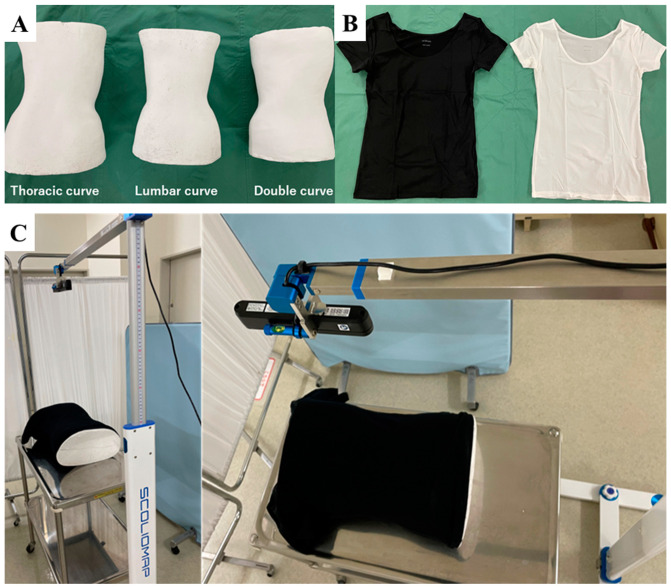
(**A**) Three different phantom models were obtained from plaster wrap castings to make hard braces for single thoracic curve, single thoracolumbar/lumbar curve, and double thoracic and double thoracolumbar/lumbar curve. (**B**) Black and white underwear types were used. (**C**). A shooting scene.

**Figure 4 jcm-12-00499-f004:**
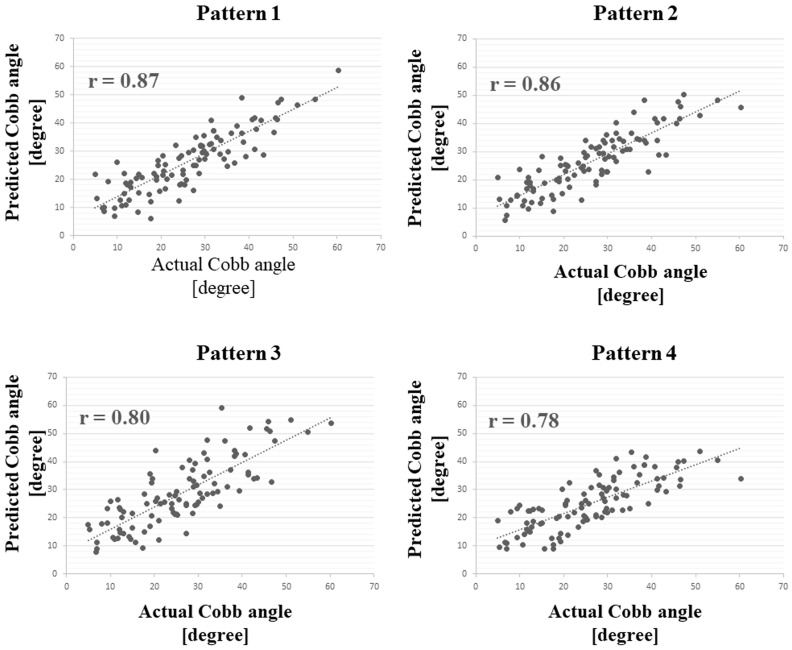
Correlation in the total number of subjects between the actual Cobb angle and the predicted Cobb angle.

**Table 1 jcm-12-00499-t001:** Predicted Cobb angle and coefficient of variation in three black underwear types.

	UNIQLO	Ryohin Keikaku	SHIMAMURA	*p* Value
Thoracic Single Curve				
Predicted Cobb angle (°)	24.8 ± 0.60	24.7 ± 0.47	24.6 ± 0.33	0.831
Coefficient of variation (%)	2	2	1	
Thoracolumbar/lumbar single curve				
Predicted Cobb angle (°)	24.0 ± 0.59	24.6 ± 1.06	23.8 ± 0.63	0.151
Coefficient of variation (%)	2	4	3	
Thoracic Thoracolumbar/lumbar double curve				
Predicted Cobb angle (°)	31.1 ± 0.56	30.9 ± 0.93	31.6 ± 0.62	0.108
Coefficient of variation (%)	2	3	2	

The values of the predicted Cobb angle are given as the average and the standard deviation.

**Table 2 jcm-12-00499-t002:** Predicted Cobb angle and coefficient of variation in two color underwear types.

	UNIQLO Black	UNIQLO White	*p* Value
Thoracic single curve			
Predicted Cobb angle (°)	24.8 ± 0.60	25.5 ± 1.09	0.098
Coefficient of variation (%)	2	4	
Thoracolumbar/lumbar single curve			
Predicted Cobb angle (°)	24.0 ± 0.59	23.6 ± 0.72	0.115
Coefficient of variation (%)	2	3	
Thoracic Thoracolumbar/lumbar double curve			
Predicted Cobb angle (°)	31.1 ± 0.53	32.3 ± 1.00	0.997
Coefficient of variation (%)		1	

The values of the predicted Cobb angle are given as the average and the standard deviation.

**Table 3 jcm-12-00499-t003:** The MAE and RMSE in each shooting pattern.

	Pattern 1		Pattern 2		Pattern 3		Pattern 4	
	MAE (°)	RMSE (°)	MAE (°)	RMSE (°)	MAE (°)	RMSE (°)	MAE (°)	RMSE (°)
Mild group (0° to 19°)	5.1	7.4	4.9	6.3	6.4	8.6	6.1	7.5
Moderate group (20° to 39°)	4.4	6.0	4.2	5.4	6.7	7.8	4.7	5.8
Severe group (≥40°)	4.7	7.1	6.0	7.7	6.2	7.2	11.0	12.0
Total	4.7	6.0	4.8	6.1	6.3	8.0	6.1	7.6

MAE = Mean absolute error, RMSE = Root mean square error.

**Table 4 jcm-12-00499-t004:** *p*-value of the Tukey post hoc test between the two MAEs.

	Pattern 1	Pattern 2	Pattern 3	Pattern 4
Pattern 1	N/A	0.99	0.03	0.09
Pattern 2		N/A	0.04	0.11
Pattern 3			N/A	0.97
Pattern 4				N/A

MAE = Mean absolute error, N/A = not available.

**Table 5 jcm-12-00499-t005:** Experimental indicators of the averaged predicted Cobb angles.

Pattern	Cobb Angle	Sensitivity	Specificity	PPV	NPV	Accuracy	PLR	NLR
1	10°	0.98	0.36	0.93	0.67	0.92	1.54	0.06
	15°	0.94	0.52	0.88	0.69	0.86	1.97	0.12
	20°	0.89	0.69	0.84	0.77	0.83	2.84	0.16
	25°	0.83	0.85	0.86	0.82	0.85	5.57	0.20
2	10°	0.98	0.18	0.89	0.50	0.89	1.19	0.13
	15°	0.92	0.52	0.88	0.65	0.85	1.94	0.14
	20°	0.94	0.71	0.86	0.86	0.87	3.28	0.09
	25°	0.85	0.85	0.87	0.83	0.86	5.70	0.18
3	10°	0.99	0.18	0.91	0.67	0.91	1.21	0.06
	15°	0.94	0.43	0.86	0.64	0.84	1.64	0.15
	20°	0.95	0.57	0.81	0.87	0.83	2.23	0.08
	25°	0.83	0.70	0.76	0.79	0.78	2.79	0.24
4	10°	0.98	0.18	0.91	0.50	0.90	1.19	0.12
	15°	0.89	0.33	0.83	0.44	0.78	1.32	0.34
	20°	0.94	0.69	0.85	0.86	0.86	2.99	0.09
	25°	0.77	0.87	0.87	0.77	0.83	6.06	0.26

PPV = positive predictive value, NPV = negative predictive value, PLR = positive likelihood ratio, NLR = negative likelihood ratio.

## Data Availability

The data that support the findings of this study are available from the corresponding author on reasonable request.

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
