# Peer review of "Prediction of Cobb Angle Using Deep Learning Algorithm with Three-Dimensional Depth Sensor Considering the Influence of Garment in Idiopathic Scoliosis"

_jcm, 2023, doi:10.3390/jcm12020499_

Round 1

Reviewer 1 Report

Dear Authors,

This paper discusses very important research area regarding Idiopathic Adolescent Scoliosis and Cobb angle measurements in young people.  The paper is well structured and well written, but some recommendations could improve its quality. I hope my comments and minor revisions would be helpful for improvement of the manuscript, before the manuscript could be considered for publication. Please find below my comments for the sections of the paper.

1.There is an absolutely the same text:

in the Rows 231-233 (in Discussion) and in 256-258 (in Conclusions). Please do changes or delete part of it.

2.The same text is in the Rows 225-228 and in 252-256. Please, avoid repetitions. Please, check the whole text carefully, to avoid other possible repetitions.

3. In Row 246, it is unclear, to whom is addressed this sentence. We know, as normally the paper is addressed to the readers. It is misleading paragraph, and I would suggest to consider, do you really need it at all (maybe it could be deleted?).

4.In Discussion section: In the beginning of the Discussion section it is recommended firstly to say shortly about the main findings of the study, and after you compare your results with the findings of other studies and authors.

5. There are not many references (11) used in your study, among them – 8 are new, and 3 older. Is it because of the specific of the paper, or some other reasons?

6. In Row 134, please correct cobb angles into Cobb angles (use Cobb from capital letter). I am sure it is just a technical error.

Please, highlight the changes to the revised version, using a different colour.

Happy New Year!

Reviewer 2 Report

This is an interesting study on scoliosis patients.

I have no major concerns; the approach seems sound and the results are consistent. Nevertheless, I think some aspects could be improved:

Methods:

1) "All methods were performed in accordance with the relevant guidelines and regulations" - which guidelines? which regulations? Please describe.
